# Investigation of the Impact of Lipid Acyl Chain Saturation on Fusion Peptide Interactions with Lipid Bilayers

**William T. Heller *** and **Piotr A. Zolnierczuk**

Neutron Scattering Division, Oak Ridge National Laboratory, Oak Ridge, TN 37831, USA
* Correspondence: hellerwt@ornl.gov; Tel.: +1-865-241-0093

**Abstract:** The interaction of many peptides with lipid bilayer membranes strongly depends on the lipid composition. Here, a study of the impact of unsaturated lipid acyl chains on the interaction of a derivative of the HIV-1 fusion peptide with lipid bilayer vesicles is presented. Lipid bilayer vesicles composed of mixtures of lipids with two saturated acyl chains and lipids and one saturated and one unsaturated acyl chain, but identical head groups, were studied. The dependence of the peptide conformation on the unsaturated lipid content was probed by circular dichroism spectroscopy, while the impact of the peptide on the bilayer structure was determined by small-angle neutron scattering. The impact of the peptide on the lipid bilayer vesicle dynamics was investigated using neutron spin echo spectroscopy. Molecular dynamics simulations were used to characterize the behavior of the systems studied to determine if there were clear differences in their physical properties. The results reveal that the peptide–bilayer interaction is not a simple function of the unsaturated lipid acyl chain content of the bilayer. Instead, the peptide behavior is more consistent with that seen for the bilayer containing only unsaturated lipids, which is supported by lipid-specific interactions revealed by the simulations.

**Keywords:** fusion peptide; lipid bilayer; neutron spin echo spectroscopy; small-angle neutron scattering; molecular dynamics simulations

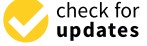



## 1. Introduction

An interesting feature of the behavior of the fusion peptide (FP) of HIV-1, which is the 23-residue N-terminal segment of the gp41 glycoprotein of the virus [1], is that its conformation depends on the lipid bilayer that it is associated with and its concentration. The wild-type peptide (amino acid sequence: AVGIGALFLGFLGAAGSTMGARS) and mutations that retain the bulk of this sequence adopt a conformation that is a mixture of α-helix and random coil when it is associated with neutral lipid bilayers or when present at lower concentrations [2–7]. Higher concentrations and lipid bilayers containing charged lipids favor a conformation that contains a significant fraction of β-sheet structure [2–4,6–8], which is considered to be the fusion-promoting state of the FP. These broad statements could lead to the conclusion that the polar head group regions of the lipid bilayer membrane are the sole determinants of how the FP behaves. The hydrocarbon core of the bilayer also plays a role in controlling the behavior of the FP. At comparable peptide concentrations, the β-sheet conformation is more likely to be adopted in lipid bilayers with saturated chains [6,7,9–11] than it is when the chains are unsaturated [12,13]. Additionally, cholesterol can promote the β-sheet conformation of the FP and fusion [9–26].

Neither the hydrocarbon core of the bilayer nor the polar head group region function independently of the other, but the hydrocarbon core provides the hydrophobic barrier between the interior and exterior of the cell. The hydrocarbon region of a lipid bilayer possesses several physical characteristics, such as thickness, order, packing and hydrophobicity, that arise from its composition and impact its properties. Unsaturated bonds increase the disorder of the acyl chains [27–30] and produce a more hydrophobic lipid bilayer

core [31,32]. Cholesterol makes the core of the bilayer more hydrophobic, and it significantly alters the relative hydrophobicity of the different regions of the bilayer [31,32]. It also causes the acyl chains to become more ordered and pack more tightly [33,34], which is more pronounced when both acyl chains are fully saturated [34]. Cholesterol also makes bilayers more rigid, but to a lesser extent when one chain is unsaturated than when both are saturated [34–38].

The difference in peptide behavior between lipid bilayers with and without unsaturated acyl chains raises questions about the sensitivity of the peptide to the acyl chain content of the lipid bilayer. Here, we investigated the interaction of gp41rk, a variant of the gp41 FP that has the sequence RKGIGALFLGFLGAAGSTMKR-NH$_2$ [6,7,13], with lipid bilayer vesicles composed of 1,2-dimyristoyl lipids, where both acyl chains of the lipid are fully saturated, with 1-palmitoyl-2-oleyl lipids, which have one acyl chain fully saturated, and the other having a single unsaturated bond, to better understand how lipid acyl chain composition impacts the interaction. The lipid headgroup composition was kept constant. CD spectroscopy, SANS and NSE spectroscopy were used to probe conformation, bilayer structure and vesicle dynamics. These studies were complemented with MD simulations to better understand what is taking place at the molecular level. The results provide new insight into the interaction of the peptide with lipid bilayer vesicles.

## 2. Materials and Methods

### 2.1. Materials

The amidated form of the gp41rk peptide was synthesized by Biomatik USA, LLC (Wilmington, DE, USA). The peptide purity was 98.2%, based on the manufacturer's HPLC analysis (see Figure S1). It was used without further purification. The lipids DMPC, DMPG, POPC and POPG were purchased from Avanti Polar Lipids (Alabaster, AL, USA) and were in either chloroform solution or in powder form. All lipids were used without further purification. Chloroform (Amresco, LLC, Solon, OH, USA), TFE (Fisher Scientific, Pittsburgh, PA, USA) and D$_2$O (Cambridge Isotope Laboratories, Inc., Tewksbury, MA, USA) were also used without further purification. A stock solution of the peptide was prepared in TFE at 10 mg/mL to simplify the sample preparation.

### 2.2. Vesicle Preparation

Sample preparation followed the procedure employed in previous studies of gp41rk [6,7,13]. Briefly, appropriate quantities of the lipids and peptides were added to a vial and approximately 1 mL of a 3:1 chloroform:TFE mixture was added to ensure that the lipids and peptides mixed at the desired molar ratios, denoted P/L. Samples with P/L = 0 and 1/50 were prepared for the experiments. Three lipid mixtures, which are also molar mixtures, were investigated: 7:3 DMPC:DPMG, 7:3 POPC:POPG and 1:1 (7:3 DMPC:DMPG):(7:3 POPC:POPG), which are abbreviated as Lipid-1, Lipid-2 and Lipid-3, respectively, for the sake of brevity. All solvents were then blown off under a stream of dry N$_2$ before the samples were placed in a vacuum freeze-drier for at least 12 h to remove any remaining solvent. The samples were then re-suspended in D$_2$O at a lipid concentration of 5 mg/mL for the SANS and CD experiments and 30 mg/mL for the NSE experiment. All samples were vortexed extensively to ensure that all material went into the solution. The resulting solutions were subjected to three freeze–thaw cycles between −80 and 40 °C. The samples were vortexed after each thaw cycle. Finally, the solutions were extruded using an Avanti Polar Lipids mini-extruder (Alabaster, AL, USA) fit with a 100 nm pore polycarbonate membrane. The samples were passed through the membrane at least 21 times before being placed in a 12 °C incubator until the start of the SANS or NSE measurements the following day. The CD spectroscopy measurements used aliquots of the SANS samples.

### 2.3. CD Spectroscopy

The peptide conformation was characterized using circular dichroism (CD) spectroscopy. All samples were measured with a Jasco J-810 CD spectropolarimeter (JASCO International Co., Ltd., Tokyo, Japan). The sample temperature was 37 °C during the measurements. Vesicle solutions, which were also the SANS samples and were not diluted, were loaded into a 0.1 mm path-length quartz cuvette with demountable windows (model 106-0.10-40) from Hellma (Müllheim, Germany). CD data were collected from 180 to 260 nm in 0.2 nm steps. Ten spectra were averaged for each sample and background pair, being the gp41rk-containing and peptide-free vesicle samples. The peptide-free vesicle CD data were subtracted from those with gp41rk to correct for any background resulting from the lipids. Then, the resulting spectra were converted into the mean residue ellipticity, which has units of mdeg cm$^2$ dmol$^{-1}$, using the peptide concentration, path length and number of amino acid residues to produce the final result. The secondary structures of gp41rk in the various lipids were estimated from the CD data with the BeStSel software [39,40].

### 2.4. Small-Angle Neutron Scattering Experiments

Small-angle neutron scattering (SANS) data were collected using the EQ-SANS instrument at the Spallation Neutron Source of Oak Ridge National Laboratory [41]. Data were collected using one sample-to-detector distance, namely 4 m, and two minimum wavelength settings, specifically 10 and 2.5 Å. The temperature was maintained at 37 °C during the experiment. All samples were measured in 1 mm path-length cylindrical quartz cells from Hellma (Müllheim, Germany). In addition to the samples noted above, SANS data were collected for an empty cell and D$_2$O. Data reduction against the empty cell followed standard procedures implemented in drtsans [42], which uses algorithms from Mantid [43] for some operations. Absolute intensity scaling into units of cm$^{-1}$ was accomplished through the use of a calibrated standard [44]. The samples were isotropic, so the 1D $I(q)$ vs. $q$ data were used during the data analysis, where $q = 4\pi sin(\theta)/\lambda$ is the magnitude of the momentum transfer, $2\theta$ is the scattering angle, and $\lambda$ is the neutron wavelength.

### 2.5. Small-Angle Neutron Scattering Data Analysis

The SANS data were analyzed using a three-shell model that was implemented in Python. Data fitting was performed with the *sas-temper* software [45]. A full description of the model is provided in the Supplementary Materials, along with a model schematic (Figure S2) and the scattering length densities (Table S1). Briefly, it is a three-layer vesicle model, and the layer thicknesses and compositions were determined from the volumes of the chemical groups of the lipids and the average area per lipid, $A_L$, which was one free parameter in the model. The peptide was considered, if it was present, and it was assumed to be in one of two states: inserted across the hydrocarbon core or associated with the lipid headgroup region of the bilayer. The fraction of the peptide inserted into the bilayer, $f_{insert}$, was another free parameter. However, it was assumed that all peptides were bound to the vesicles, rather than there being a population free in solution, based on a previous study of the wild-type peptide that found binding at a level of ~90% when charged lipids are present [16]. If only 90% of gp41rk is bound, the effective P/L would be ~1/56 and would have a limited impact on the scattering length densities of the various layers in the model. The model had two additional free parameters: a multiplicative scale factor and a constant baseline. Some parameters, such as the thickness of the lipid headgroup region, were set to fixed values based on previously published work [46–52]. The $q$-range that was fit during data analysis was restricted to 0.01 Å$^{-1}$ ≤ $q$ ≤ 0.35 Å$^{-1}$ to avoid issues at high and low $q$ that can arise from inelastic scattering [53]. For each SANS data set, the *sas-temper* software [45] was used to generate 25 independent models.

### 2.6. Neutron Spin Echo Spectroscopy

Neutron spin echo (NSE) data were collected using the SNS-NSE spectrometer at the Spallation Neutron Source of Oak Ridge National Laboratory [54]. The measurements

covered a momentum transfer range of $0.05\,\text{Å}^{-1} \leq q \leq 0.12\,\text{Å}^{-1}$, and the time range was $0.03\,\text{ns} \leq \tau \leq 100\,\text{ns}$. During the measurements, the temperature was maintained at 37 °C using a ThermoJet ES system (SP Scientific, Stony Ridge, NY, USA). The samples were loaded into rectangular (3 mm path length, $30 \times 30\,\text{mm}^2$ cross-section) quartz cells from Hellma (Müllheim, Germany). Measurements using elastically scattering reference samples (graphite foils and $Al_2O_3$) and $D_2O$ were also taken in order to allow for instrument resolution correction and background subtraction. The data were reduced with the program *DrSpine* [55] to yield the resolution and background-corrected dynamic structure factor $S(q, \tau)$.

*2.7. Neutron Spin Echo Spectroscopy Data Analysis*

The effective bending modulus of the bilayers was extracted from the dynamic structure factor by fitting the data using the Zilman–Granek model [56,57] shown in Equation (1).

$$S(q, \tau) = S(q, 0)e^{-(\Gamma(q)\tau)^{2/3}}. \tag{1}$$

The decay rate $\Gamma(q)$ in Equation (1) is given by Equation (2).

$$\Gamma(q) = 0.025\gamma\sqrt{\frac{kT}{\tilde{\kappa}}}\frac{kT}{\eta_{D_2O}}q^3. \tag{2}$$

The viscosity of $D_2O$, $\eta_{D_2O}$ at 37 °C is 0.837 cP [58], $k_B$ is the Boltzmann constant, and $T$ is the temperature in Kelvin. The parameter $\gamma$ is a weakly temperature-dependent parameter [56], but it was assumed to be unity in previous studies [7,59–61]. In Equation (2), $\tilde{\kappa}$ is the effective bending modulus. $\tilde{\kappa}$ can be used to derive the bending modulus $\kappa$ and the compressibility modulus $k_m$ [57,59,62]. Doing so requires the use of parameters that do not have well-known values, which results in values of $\kappa$ and $k_m$ that are best described using ranges [7]. The values of the effective bending moduli with and without peptide are instructive and sufficient for describing the relative impact of gp41rk on the various bilayers studied here. A similar approach was recently used for interpreting NSE data from lipid bilayers interacting with other peptides [63].

*2.8. Molecular Dynamics Simulations*

Molecular dynamics simulations of the peptide-free lipid mixtures, as well as one gp41rk interacting with a Lipid-3 bilayer, referred to as gp41rk/Lipid-3, were performed using GROMACS [64], version 2021.4 [65]. The Stockholm lipids force field parameters were used for the simulations [66–70], and it was used with the AMBER99SB-ILDN force field [71] in the Lipid-3/gp41rk simulation because the two force fields are compatible. Membranes containing 200 lipids and either 50 water molecules per lipid (Lipid-1) or 40 water molecules per lipid (Lipid-2 and Lipid-3) were constructed for Lipid-1, Lipid-2 and Lipid-3 using the MemGen web server [72]. The reduced number of waters per lipid improved the computational speed of the MD simulations. Both levels of hydration were used represent a fully hydrated system. Sodium counter ions were included with the constructed bilayers. Both leaflets of the bilayer have identical compositions [72]. The starting structure for gp41rk was one of a set produced using I-TASSER [73,74]. The starting conformation selected from the set had a random coil conformation. The peptide was initially placed in the water layer as near to the bilayer as possible while not allowing collisions using a simple software tool developed in-house. A random coil conformation was selected because the peptide adopts this conformation in water [6]. The tool identified water molecules displaced by the protein and those between the bilayer surface and gp41rk and then removed those water molecules. The water removal did not significantly alter the level of hydration of the system.

The parameters for the various MD simulations performed are as follows. Full 3D periodic boundary conditions were employed in all cases. First, an energy minimization step was run that used the steepest descent minimization algorithm, a step size of 0.01 nm, and a

target maximum force of 500 kJ/mol/nm when no peptide was present and 250 kJ/mol/nm when the peptide was present to reduce the possibility of problems in the more complicated system. A 1.2 nm cut-off was used for both electrostatics, performed using the particle mesh Ewald (PME) approach [75], and van der Waals calculations. Up to 50,000 steps were allowed but were not required for any of the simulations. Then, a short (100 ps) NVT ensemble simulation was performed. The leap-frog integrator was employed with a time step of 2 fs. PME [75] was again used for electrostatics with a 1.2 nm cut-off distance, and the same cut-off distance was applied to the van der Waals calculations. The temperature was maintained using velocity rescaling [76] with a time constant of 0.1 ps. Two temperature coupling groups, the lipids and the water + ions, were used in the peptide-free simulations, while a third group containing the peptide was added for the gp41rk/Lipid-3 simulation because of the expectation that the peptide would transition from the water to interacting with the bilayer.

After the NVT ensemble simulation was complete, runs were performed using the NPT ensemble. All NPT simulations used the same parameters, described below. For the peptide-free simulations, a 1 ns equilibration run was performed and inspected prior to the production runs, while a 10 ns equilibration simulation was performed for the gp41rk/Lipid-3 system and inspected. A 200 ns production run was performed for the peptide-free systems. In the case of the gp41rk/Lipid-3 system, four 100 ns production runs were performed to allow the peptide to bind to the bilayer and settle, which took place within ~200 ns of total simulation time. After confirming that binding had taken place and that this state was maintained for 100 ns by comparing the lipid phosphate density profile with that of the peptide, a final 200 ns long production run was performed. Electrostatic and van der Waals calculations used the same parameters as the NVT simulations. The Nosé-Hoover thermostat [77,78] with a time constant of 0.5 ps was used to maintain a temperature of 37 °C (310 K). The temperature coupling groups used in the NVT ensemble simulations were used in the NPT ensemble simulations. To maintain the pressure, Parrinello–Rahman semi-isotropic pressure coupling [79] was used with a time constant of 5 ps to maintain a pressure of 1 bar.

The simulation results were analyzed using the tools included with GROMACS [64]. The density of the system along the direction normal to the bilayer was calculated using the *density* tool. RDFs were calculated using the *rdf* tool, and a 60 Å maximum distance was employed. MSDs and RMSFs were calculated using the *rms* and *rmsf* tools, respectively, and the default fitting parameters were used for determining the diffusion coefficients from the MSD plots. The secondary structure of the protein was calculated using the DSSP program [80,81] using the *do_dssp* tool. Lipid acyl chain order parameters were calculated using the *order* tool. The minimum distances between the amino acids of gp41rk and the bilayer were evaluated with *mindist*.

### 3. Results

#### *3.1. CD*

The CD spectra collected from the peptide in the three lipid mixtures studied are shown in Figure 1. The data collected show some evidence of attenuation artifacts and noise below 185 nm, but these wavelengths are below the main spectral features. In Lipid-1, the spectrum has a weak negative peak near 220 nm and a single positive peak just below 200 nm, consistent with previous work using fully saturated lipid mixtures [6,7]. The spectra in Lipid-2 and Lipid-3 have negative peaks near 220 and 208 nm, and a positive peak near 190 nn, which is also consistent with previous studies of gp41rk in unsaturated lipids [13]. The secondary structure content of the peptide was determined using the *BeStSel* software [39,40], and the results are presented in Table 1. The peptide conformation in all lipid mixtures contains considerable "other" secondary structures, being structures that cannot be classified as α-helix, β-sheet or turns. Importantly, the peptide conformation in both Lipid-2 and Lipid-3 contains considerably more α-helical content than β-sheet and turns. The data collected for the peptides in Lipid-3 are not consistent with an equally

weighted mixture of the data from Lipid-1 and Lipid-2. The spectrum calculated as an equally weighted mixture of these two data sets are the green curve in Figure 1.

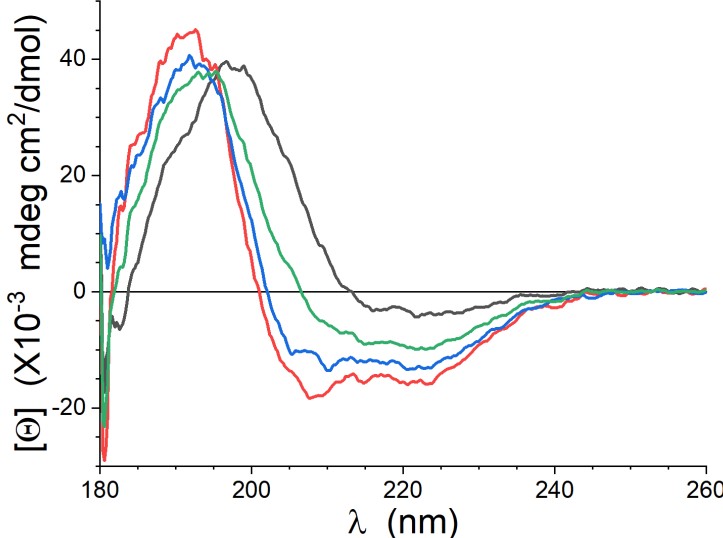

**Figure 1.** CD data of gp41rk in Lipid-1 (black), Lipid-2 (red) and Lipid-3 (blue) at P/L = 1/50. The green curve is an equally-weighted superposition of the Lipid-1 and Lipid-2 data (i.e., $[\Theta]_{mix}(\lambda) = 0.5[\Theta]_{Lipid-1}(\lambda) + 0.5[\Theta]_{Lipid-2}(\lambda)$.

**Table 1.** Results of secondary structure determination from the CD spectra using BeStSel [39,40].

| Lipid | $\alpha$-Helix | Anti-Parallel $\beta$-Sheet | Turns | "Other" |
|---|---|---|---|---|
| Lipid-1 | 5% | 61% | 14% | 20% |
| Lipid-2 | 40% | 5% | 5% | 50% |
| Lipid-3 | 42% | 11% | 13% | 34% |

*3.2. SANS*

The SANS data collected from the peptide-free vesicles and those with gp41rk present at P/L = 1/50 are shown in Figure 2. The differences between the various data sets are rather subtle, as was observed previously [7]. Importantly, the SANS data do not show signs of multilamellar vesicles, such as a peak, knee or shoulder near $q = 0.1$ Å$^{-1}$ that arises from multilamellar diffraction. The power-law slope for $q < 0.05$ Å$^{-1}$ is also consistent with a core-multi-shell model's expected slope of $-2$. The presence of a feature near $q = 0.1$ Å$^{-1}$ and a low-$q$ power-law slope less than $-2$ are indicators of multilamellar vesicles and would indicate fusion [82]. Both of these indications of fusion were previously observed by SANS when gp41rk was associated with 7:3 POPC:1-palmitoyl-2-oleoyl-sn-glycero-3-phospho-L-serine (sodium salt) (POPS) lipid bilayer vesicles containing 30 mol% cholesterol but not when less cholesterol was present in the mixture [13].

Figure 2 also includes example model intensity profiles resulting from the SANS data analysis. The results of the data analysis are presented in Table 2. More detailed information, including the range of $\chi^2$ values, is provided in Table S2. The fits of the models to the data are good, as can be seen in Figure 2. The data analysis shows subtle changes in the bilayer structure with the inclusion of peptides. $A_L$ increases in response to gp41rk but to a greater extent for Lipid-2 and Lipid-3 than for Lipid-1. Neither $D_{hc}$ nor $D_b$ change significantly for Lipid-1. $D_b$ decreases for Lipid-2 and Lipid-3, but only Lipid-2 displays a considerable decrease in $D_{hc}$. The averages and standard deviations of $f_{insert}$ shown in Table 2 do not suggest that there is a clear trend as a function of lipids, or a strong indication of a preference for the peptide to be inserted across the bilayer. However, plots

of $f_{insert}$ vs. $\chi^2$, which are output by *sas-temper* [45] and are presented in Figure S3, indicate that the Lipid-1 at P/L = 1/50 SANS data is fit better when $f_{insert}$ is higher, while the SANS data from Lipid-2 and Lipid-3 at P/L = 1/50 fit better when $f_{insert}$ is lower. Overall, the impact of the peptide on Lipid-3 is intermediate to the impact of it on Lipid-1 and Lipid-2.

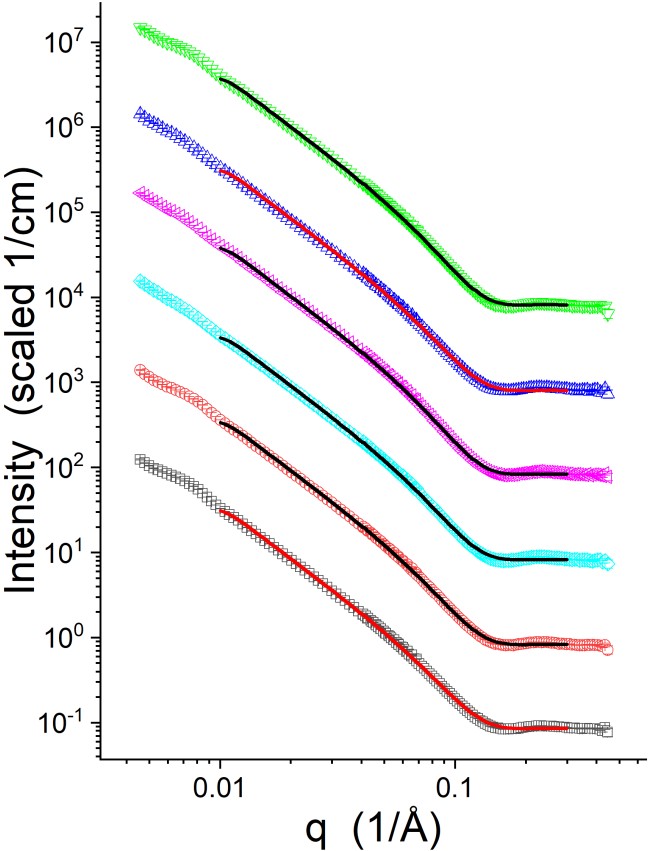

**Figure 2.** SANS data collected for the vesicles with and without gp41rk. The curves are Lipid-1 P/L = 0 (□), Lipid-1 P/L = 1/50 (○), Lipid-2 P/L = 0 (◇), Lipid-2 P/L = 1/50 (◁), Lipid-3 P/L = 0 (△), and Lipid-3 (▽). Example model intensity profiles found during the SANS data analysis are shown plotted as solid lines with the various data sets in colors that contrast sufficiently with the data. The data have been offset from Lipid-1 at P/L = 0 data for clarity.

**Table 2.** Results of the SANS data modeling highlighting the important structural parameters. The $A_L$, $f_{insert}$ that were determined by fitting the SANS data using the model described in the Supporting Information. The parameters derived from these two free parameters, $D_{hc}$, $D_b$ and $n_w$, are also presented. All four free parameters are presented in Table S2. The values of $A_L$ and $f_{insert}$ presented here are the averages and standard deviations that were provided by *sas_temper* [45] (Section 2.5). Uncertainties in the derived parameters were propagated from the standard deviations in the fit parameters.

| Sample | $A_L$ (Å$^2$) | $f_{insert}$ | $D_{hc}$ (Å) | $D_b$ (Å) | $n_w$ |
|---|---|---|---|---|---|
| Lipid-1, P/L = 0 | 59.2 ± 0.1 | 0 | 13.2 ± 0.1 | 44.4 ± 0.2 | 7.3 ± 0.1 |
| Lipid-1, P/L = 1/50 | 59.7 ± 0.1 | 0.48 ± 0.24 | 13.1 ± 0.1 | 44.1 ± 0.2 | 6.6 ± 0.4 |
| Lipid-2, P/L = 0 | 65.1 ± 0.1 | 0 | 14.4 ± 0.1 | 46.8 ± 0.2 | 9.1 ± 0.1 |
| Lipid-2, P/L = 1/50 | 66.8 ± 0.1 | 0.35 ± 0.19 | 14.0 ± 0.1 | 46.0 ± 0.2 | 8.9 ± 0.4 |
| Lipid-3, P/L = 0 | 62.9 ± 0.1 | 0 | 13.6 ± 0.1 | 45.3 ± 0.2 | 8.4 ± 0.1 |
| Lipid-3, P/L = 1/50 | 63.7 ± 0.1 | 0.45 ± 0.27 | 13.5 ± 0.1 | 44.9 ± 0.2 | 7.8 ± 0.5 |

### 3.3. NSE

Figure S4 presents $S(q, \tau)/S(q, 0)$ vs. $\tau$ from the NSE measurements. $\Gamma(q)/q^3$ vs. $q$ is shown in Figure 3 using $\Gamma(q)$ extracted from the NSE data using Equation (1). The average values are shown as the horizontal lines in Figure 3 and are also presented in Table 3. Note that the time available for the experiment did not permit the measurement of Lipid-1 at P/L = 0 in the present study, but the sample was studied previously [7]. The data from the previous study [7] were re-analyzed with an improved version of the reduction software [55] and are presented here. The present results are consistent with our previous findings within the uncertainties. In the case of the peptide-free vesicles, the fully unsaturated Lipid-2 mixture is slightly more rigid than the peptide-free Lipid-1 or Lipid-3 samples, although the difference is not strictly significant in light of the experimental uncertainties. The difference can be attributed to the dependence of the membrane rigidity on the bilayer thickness [57,59,62], which is greater for Lipid-2 than the other two lipids (Table 2). The inclusion of gp41rk with the Lipid-1 vesicles resulted in a value of $\Gamma(q)/q^3$ consistent with the previously measured value [7]. The peptide-driven decrease in $\Gamma(q)/q^3$ is $\sim$2 Å$^3$/ns. The decrease in $\Gamma(q)/q^3$ seen for the Lipid-2 and Lipid-3 vesicles as a result of gp41rk was $\sim$1 Å$^3$/ns. In all cases, the slower dynamics observed when the peptide was bound, which indicates stiffer vesicles, can be partially attributed to the decrease in the net charge of the vesicles [83,84].

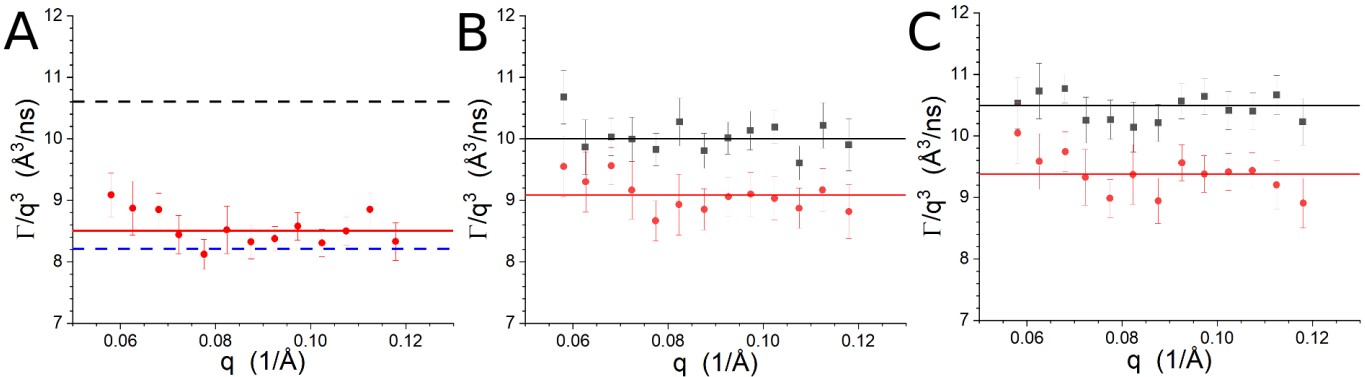

**Figure 3.** Relaxation rates divided by $q^3$ as a function of $q$ for Lipid-1 (**A**), Lipid-2 (**B**) and Lipid-3 (**C**). The symbols for the samples with P/L = 0 are ■, while those for P/L = 1/50 are ●. The solid lines are the averages reported in Table 3 and have the same color as the symbols. The dashed lines in (**A**) are the average values for P/L = 0 (black) and P/L = 1/50 (blue) determined in the previous study of gp41rk associated with Lipid-1 [7].

**Table 3.** Relaxation rates, $\Gamma(q)/q^3$, and effective bending moduli, $\tilde{\kappa}$, derived from the NSE results.

| Sample | $\Gamma(q)/q^3$ (³/ns) | $\tilde{\kappa}/k_b T$ |
|---|---|---|
| Lipid-1, P/L = 0 from [7] | 10.6 ± 0.4 | 150 ± 10 |
| Lipid-1, P/L = 1/50 from [7] | 8.2 ± 0.3 | 240 ± 30 |
| Lipid-1, P/L = 1/50 | 8.5 ± 0.4 | 230 ± 10 |
| Lipid-2, P/L = 0 | 10.0 ± 0.2 | 160 ± 10 |
| Lipid-2, P/L = 1/50 | 9.1 ± 0.4 | 200 ± 10 |
| Lipid-3, P/L = 0 | 10.5 ± 0.2 | 150 ± 10 |
| Lipid-3, P/L = 1/50 | 9.4 ± 0.3 | 190 ± 10 |

### 3.4. MD Simulations

MD simulations were performed to understand the peptide-free bilayers and to understand how the peptide interacts with the lipids in the Lipid-3 mixture. RMSF values as a function of atom number are shown in Figure S5. The jagged appearance of the lipid RMSF curves is not related to any specific lipid, which is contained in contiguous blocks in the coordinate files. Instead, the appearance is related to the periodic boundary conditions,

which impact the lipids, but not the protein. The *rmsf* tool in GROMACS [64] does not have an option for different treatments of period boundary conditions. The MSDs calculated from the various simulations are presented in Figure S6. Significant differences are visible between some lipids, but not in every simulation. The greatest differences are between the peptide-free and Lipid-3/gp41rk simulations, which show a flattening of the curves in the middle of the time range in the plots. The peptide clearly diffuses more slowly than the lipids in the Lipid-3/gp41rk simulation, which is reasonable because of its higher molecular weight.

The mass density profiles of the simulated lipid bilayers are shown in Figure 4. The density profile of Lipid-3 is more like that of Lipid-2 than that of Lipid-1, but the differences between the three peptide-free density profiles are subtle. The $A_L$ determined from the three simulations using the dimensions of the simulation box retained as part of the simulation output are $64.0 \pm 1.4$ Å$^2$ for Lipid-1, $67.6 \pm 1.3$ Å$^2$ for Lipid-2 and $65.5 \pm 1.2$ Å$^2$ for Lipid-3. The trend of the $A_L$ for each bilayer is consistent with expectations for increasing unsaturated acyl chain content. The peptide-free simulation $A_L$ are larger and more closely spaced than the experimentally-determined $A_L$. When gp41rk is bound to the bilayer, the total area of the simulation box increases by ~2% or ~130 Å$^2$. The increase in effective $A_L$ is consistent with the trend observed in the SANS data analysis when peptides are present. Extraction of the phosphorous density profiles and fitting the peak positions reveal separations of 34.8 Å, 37.1 Å, 36.4 Å and 35.0 Å for Lipid-1, Lipid-2, Lipid-3 and gp41rk/Lipid-3, respectively. The trend in this distance is consistent with the bilayer thicknesses determined from the SANS results, although the impact of the peptide is slightly larger in the simulations. Lipid chain order parameters are presented in Figure S7 and do not indicate a dramatic decrease in the acyl chain order parameters of any of the lipids in the Lipid-3 simulation upon peptide binding. A relatively small change can be seen in the unsaturated acyl chain of POPG.

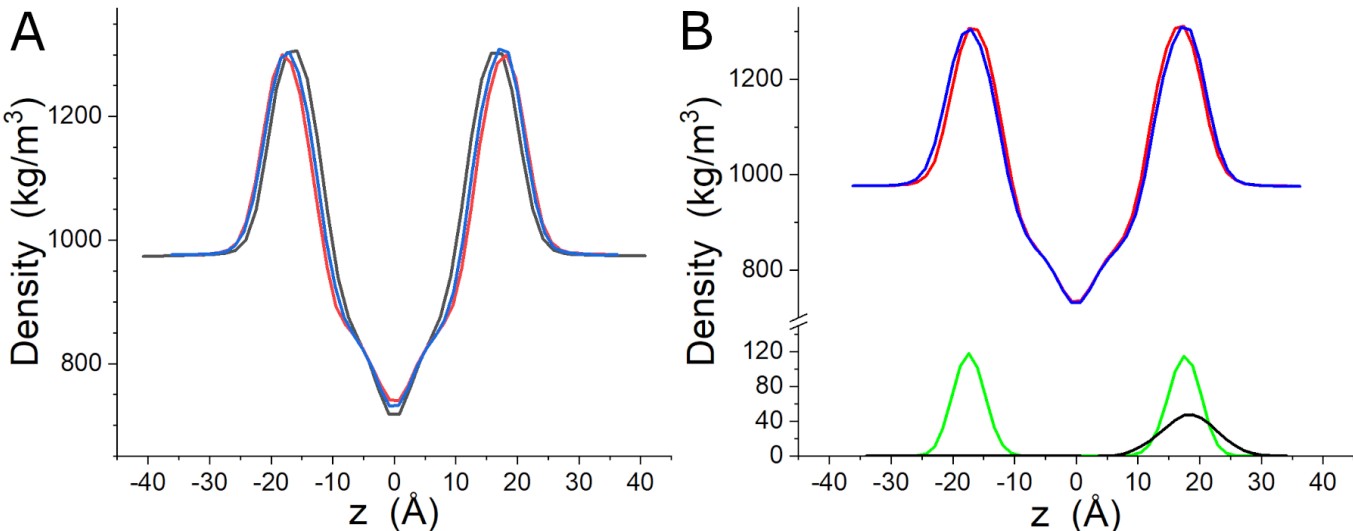

**Figure 4.** Bilayer density profiles derived from MD simulations for the peptide-free bilayers (**A**) and the Lipid-3 and gp41rk/Lipid-3 bilayers (**B**). In (**A**), the curves are Lipid-1 (black), Lipid-2 (red) and Lipid-3 (blue), while in (**B**), the curves are Lipid-3 (blue), gp41rk/Lipid-3 (red), the lipid P atoms (green) and gp41rk (black).

The phosphorous-to-phosphorous radial distribution functions (RDFs), $g(r)$, for PG lipids relative to PC lipids in the three peptide-free simulations are shown in Figure 5A,B. The general shapes of the peaks at distances less than 15 Å are very similar, and the cause of the differences in the details of the curves can be attributed to the different $A_L$ observed for each simulated bilayer. The first three peaks of the RDFs suggest that PG lipids are within the local environment (i.e., the nearest few neighbors) of PC lipids, supporting the idea that the samples are mixed rather than being phase-separated into larger domains enriched in PC and PG lipids. The features in the RDFs that are at distances greater than 30 Å are heavily influenced by distances between phosphates in the two leaflets of the bilayer.

The gp41rk $C_\alpha$ atom-to-lipid phosphorous atom RDFs are shown in Figure 5C. The RDFs for the four different lipids in the simulation were calculated for the lipids in each leaflet of the bilayer. The most striking feature of the RDFs when gp41rk is present is the stark depletion of DMPC from the vicinity of the peptide and the strong enhancement of POPG. Similarly, POPC shows a slightly stronger preference for the area around the peptide than DMPG, but the differences are much smaller than the one between the other two lipids. The peptide does not impact the distribution of lipids on the opposite leaflet of the bilayer to the same extent (Figure 5C inset).

The minimum distance between individual residues and the bilayer was evaluated. The results as a function of amino acid and time were binned into histograms and are presented in Figure S8. The N-terminal half of the sequence, as well as the last two charged residues at the C-terminus, associate more with the lipids than the remainder of the sequence. Residues 15–19 do not spend a great deal of time in close contact with the lipid bilayer during the simulation. Performing this analysis for each lipid type, which is presented in Figure S9, reveals that POPG and POPC interact more with gp41rk than DMPC or DMPG. POPC interacts with the interior of the sequence of the peptide more often than POPG, which is often in contact with the charged residues of gp41rk. DMPG interacts with portions of the N-terminal half of the peptide but has few interactions with the C-terminal half. The results indicate that the peptide interacts with all four lipid types at some point in time during the simulation, consistent with the RDFs in Figure 5.

The peptide conformation is largely unstructured based on the DSSP analysis of secondary structure [80,81], being less than 12–16% turns and having no helical content. The turn content increases from 12% to 16% over the course of the simulation. The secondary structure during the simulation is not consistent with the CD results. However, the result comes from a single trajectory rather than a macroscopic sample, and it is possible that the fractions of the secondary structure found in the measurements are population fractions rather than the secondary structure content of each peptide in the sample. It is not possible to determine which of these possibilities is what takes place in the samples measured by CD spectroscopy.

A snapshot of the HG region of one leaflet of the peptide-free bilayers can also be seen in Figure 6. Both Lipid-1 and Lipid-2 visually appear to be well-mixed, with the charged and neutral lipids being well interspersed. Lipid-3 also appears to be well-mixed, although the presence of four different lipids does make it harder to see in the image. A clear change in the distribution of lipids in the Lipid-3 bilayer can be seen when the peptide is present. The area immediately around gp41rk is enhanced in POPG, which is consistent with the RDF calculated from the entire trajectory. Similarly, the upper right corner of Figure 6D lacks POPG. The depletion of DMPC in the vicinity of the peptide is not as stark to the eye as the RDF in Figure 5 indicates.

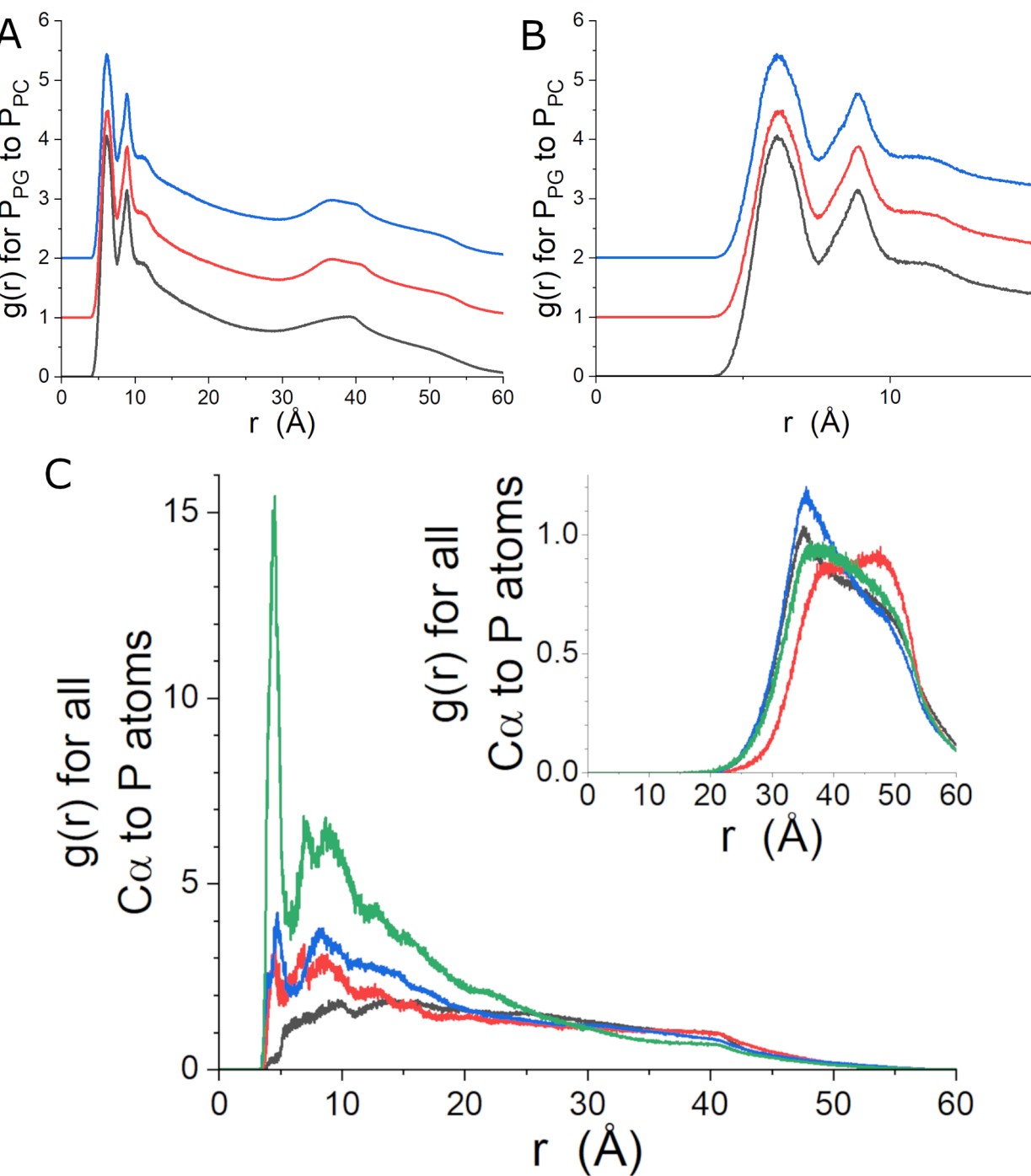

**Figure 5.** Phosphorous-to-phosphorous RDFs derived from the peptide-free MD simulations and the gp41rk/Lipid-3 simulation. The full-range (**A**) and magnified (**B**) results are shown. The curves are Lipid-1 (black), Lipid-2 (red) and Lipid-3 (blue). The curves in (**A**,**B**) have been offset for clarity. In (**C**), the RDFs for the protein $C_\alpha$ to the four lipid P atoms are shown for lipids on the same side (main panel) and opposite side (inset panel) of the bilayer as gp41rk. The curves are DMPC (black), DMPG (red), POPC (blue) and POPG (green). The relative scaling of the RDFs in all three panels is as it was provided by the GROMACS tool [64].

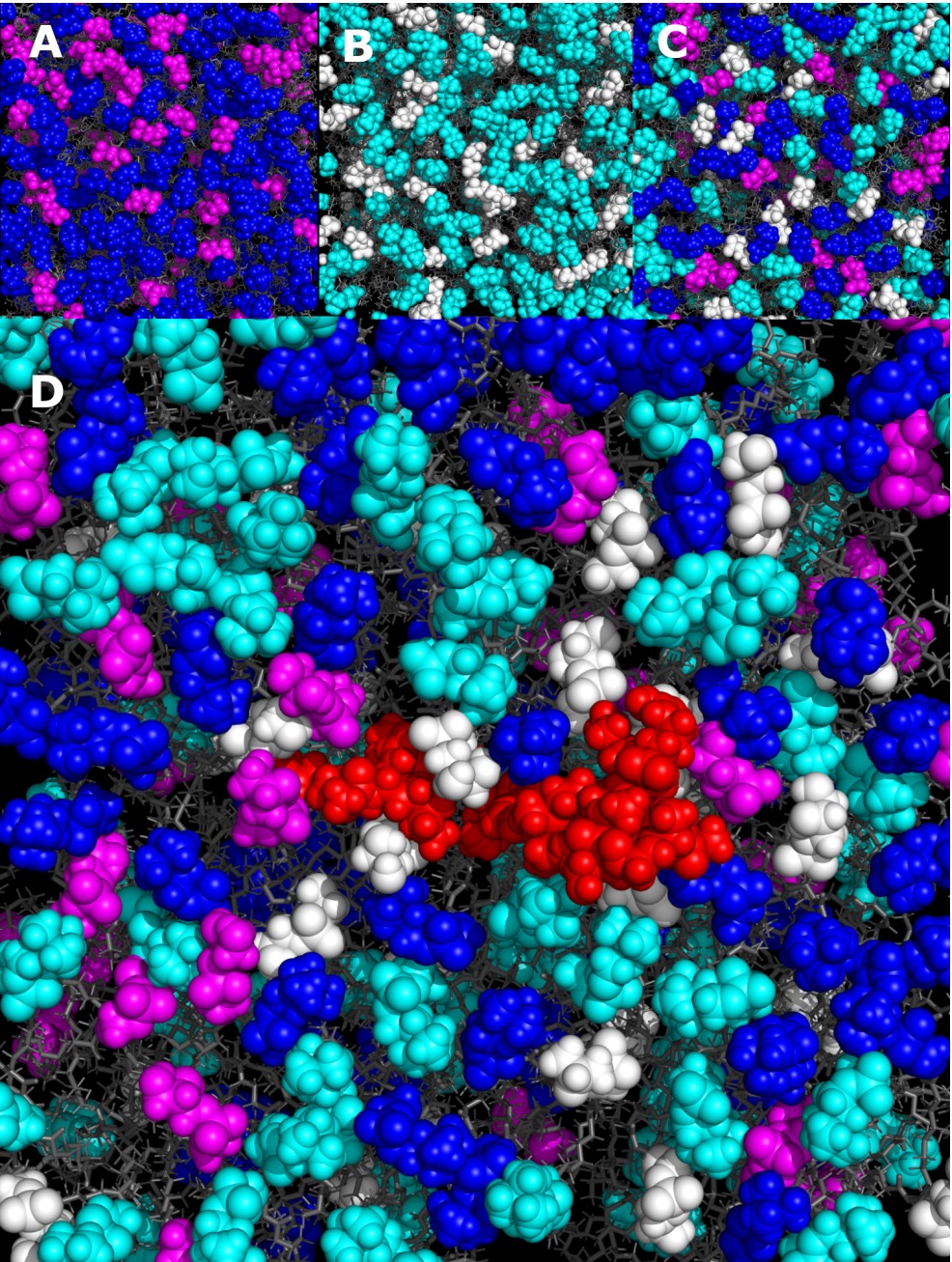

**Figure 6.** Snapshots of the MD simulations performed for (**A**) Lipid-1, (**B**) Lipid-2 and (**C**) Lipid-3 and (**D**) gp41rk/Lipid-3. Atoms within the HGs of the lipids are shown as colored spheres, while the rest of each lipid molecule is presented as grey sticks. In the images, DMPC is blue, DMPG is magenta, POPC is cyan and POPG is white. The gp41rk in (**D**) is red. The images were rendered using PyMOL [85].

## 4. Discussion

The present results reveal an interesting relationship between the lipid bilayer composition and its interaction with the FP. By studying lipid mixtures with identical head group compositions but different acyl chain compositions, a nonlinear dependence of the interaction of the peptide on the unsaturated acyl chain content was observed in the experiments performed. The MD simulations of gp41rk/Lipid-3 revealed that the area around the peptide was enhanced in POPG and depleted of DMPC. Further, POPC was slightly more prevalent in the vicinity of the peptide than DMPG. The results suggest that lipid-specific interactions take place in the Lipid-3 mixture.

The sensitivity of the conformation of the HIV-1 FP peptide and various mutations to the composition of the lipid bilayer is well-established [6,7,9–26]. The present study is unique among these studies in that only the acyl chain composition was varied while not altering the headgroup composition of the bilayer. However, it is not the first time that a nonlinear relationship between the lipid bilayer composition and fusion peptide conformation was observed. A study of the addition of cholesterol to POPC/POPG and POPC/POPS lipid bilayers on the behavior of the HIV-1 FP found that a concentration of between 20 and 30 mol% cholesterol was required before a *β*-sheet structure was observed [13–15]. Cholesterol makes the core of the bilayer more hydrophobic, and it significantly alters the relative hydrophobicity of the different regions of the bilayer [31,32] and the ordering and packing of the acyl chains [33,34] and makes bilayers more rigid [34–38]. Cholesterol is chemically and structurally very different from a phospholipid, so it is interesting that analogous behavior was observed, but the previous studies did not provide evidence of lipid-specific interactions.

It is possible to infer the mechanism that drives the lipid-specific behavior observed in the present study. The enhancement of charged lipids around the peptide can readily be attributed to electrostatic interactions. The prevalence of unsaturated acyl chains in the vicinity of the peptide can be rationalized in the following manner. The hydrocarbon region of a lipid bilayer possesses several physical characteristics, such as thickness, order, packing and hydrophobicity, which arise from its composition and impact its properties. For example, lipid bilayers with unsaturated acyl chains have a more hydrophobic core than those containing only saturated acyl chains [31,32]. Importantly, unsaturated bonds increase the disorder of the acyl chains [27–30], which creates a lipid with a more negative spontaneous curvature than lipids with saturated acyl chains. Curvature effects can influence lipid–protein interactions in a way that favors one mode of interaction over another [86]. By preferentially interacting with the head groups of POPG, and POPC to a lesser extent, gp41rk reduces the curvature strain in the system, which makes the interaction energetically favorable relative to associating with DMPC or DMPG.

The present NSE results appear to contradict previous studies of the impact of the HIV-1 FP with model lipid bilayer membranes, and it is important to address the matter. The HIV-1 FP has been studied in lipid bilayers of either pure DOPC or diC$_{22:1}$PC [87–89] using X-ray diffraction or scattering, as well as fluctuation analysis and micropipette aspiration [89]. The HIV-1 FP contains a single positively charged residue, so the softening can be partially attributed to the increased net positive charge of the bilayers [83,84]. The HIV-1 FP at P/L of 1/20 and 1/10 was also found to soften lipid bilayers composed of 10 mol% charged lipids, most of which had a single monounsaturated acyl chain [88]. The HIV-1 FP softened this more complex charged lipid mixture, even though the peptide decreased or neutralized the net charge of the system, but the effect of decreased net charge appears to have been overcome by the interaction of the peptide with the lipid bilayer at these higher concentrations.

## 5. Conclusions

A nonlinear relationship between the interaction of gp41rk with lipid mixtures having consistent lipid headgroup composition but different acyl chain compositions was found. Both the peptide conformation and how it impacts the mechanical properties of the lipid bilayer behave in this way. A similar composition dependence was seen before in mixtures of phospholipids and cholesterol [13–15]. Monounsaturated acyl chains were present in the lipid bilayer in all four studies, demonstrating their importance in the observed behavior. The MD simulations presented here indicate that the peptide preferentially associates with lipids with unsaturated acyl chains, revealing a mechanism for the observed effect.

**Supplementary Materials:** The following supporting information can be downloaded at: https://www.mdpi.com/article/10.3390/biophysica3010009/s1, Figure S1: HPLC of the gp41rk peptide; Figure S2: Schematic of the volumetric model of the lipid bilayer; Figure S3: $\chi^2$ vs. $f_{insert}$ plots; Figure S4: NSE data collected for the vesicles; RMSF values calculated from the simulations; Figure S6: MSDs calculated from the simulations; Figure S7: Lipid chain order parameters calculated from the simulations; Figure S8: Histograms of the minimum distances between the gp41rk amino acids and the lipids in the Lipid-3/gp41rk simulation; Figure S9: Histograms of the minimum distances between the gp41rk amino acids and the four types of lipids in the Lipid-3/gp41rk simulation; Table S1: Parameters for SANS data analysis; Table S2: Results of SANS data analysis.

**Author Contributions:** Conceptualization, W.T.H.; methodology, W.T.H. and P.A.Z.; formal analysis, W.T.H. and P.A.Z.; writing—original draft preparation, W.T.H. and P.A.Z.; writing—review and editing, W.T.H. and P.A.Z. All authors have read and agreed to the published version of the manuscript.

**Funding:** This research used resources at the Spallation Neutron Source, a DOE Office of Science User Facility operated by the Oak Ridge National Laboratory.

**Institutional Review Board Statement:** Not applicable.

**Informed Consent Statement:** Not applicable.

**Data Availability Statement:** Data are available from the authors on request.

**Acknowledgments:** The authors would like to thank H. M. O'Neill and K. L. Weiss for access to the CD instrument; K. L. Weiss for access to the Bio-Labs at the Spallation Neutron Source; C. Y. Gao for technical assistance with the SANS experiments and M. Odom for technical assistance with the NSE experiments.

**Conflicts of Interest:** The authors declare no conflict of interest. The funders had no role in the design of the study; in the collection, analyses, or interpretation of data; in the writing of the manuscript; or in the decision to publish the results.

## Abbreviations

The following abbreviations are used in this manuscript:

| | |
|---|---|
| FP | fusion peptide |
| SANS | small-angle neutron scattering |
| NSE | neutron spin echo spectroscopy |
| CD | circular dichroism |
| MD | molecular dynamics |
| PME | particle mesh Ewald |
| RDF | radial density function |
| RMSF | root mean square fluctuation |
| MSD | mean square displacement |
| DMPC | 1,2-dimyristoyl-sn-glycero-3-phosphocholine |
| DMPG | 1,2-dimyristoyl-sn-glycero-3-phospho-(1'-rac-glycerol) sodium salt |
| POPC | 1-palmityol-2-oleoyl-sn-glycero-3-phosphocholine |
| POPG | 1-palmityol-2-oleoyl-sn-glycero-3-phospho-(1'-rac-glycerol) sodium salt |
| POPS | 1-palmitoyl-2-oleoyl-sn-glycero-3-phospho-L-serine sodium salt |
| DOPC | 1,2-dioleoyl-sn-glycero-3-phosphocholine |
| diC$_{22:1}$PC | 1,2-dierucoyl-sn-glycero-3-phosphocholine |
| TFE | 2,2,2-trifluoroethanol |

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
