# Peer review of "Investigation of the Impact of Lipid Acyl Chain Saturation on Fusion Peptide Interactions with Lipid Bilayers"

_biophysica, doi:10.3390/biophysica3010009_

Round 1
Reviewer 1 Report
This present paper shows an extensive investigation combining experimental methods and molecular dynamics simulations in order to correlate peptide behaviour with the level of acyl chain saturation in the bilayer. The hypothesis was tested through preparation of vesicles containing three different lipid mixtures, where the headgroup composition was kept constant.
Concerning the Molecular Dynamics simulations, I'd like to read more details about the parameters used in the simulations. For example, which algorithm was used in energy minimization? Which integrator and time step adopted on MD simulations? Which values of the 'tau_p' and 'tau_t' values were adopted for control of temperature and pressure? How are the radius of non-bonded interactions as electrostatic and van der Waals?
An important question: since the peptide start the simulation in water phase, why not include it on same temperature coupling group containing water and ions?
What are the difference between the parameters used in the NPT ensemble at l 1 ns, 10 ns and 200 ns length?
Why start the simulation using a random coil conformation for the peptide, if the CD measurements determined other secondary structure for the peptide? (Table 1).
I think, most information can be extracted from MD simulations and could enrich the discussion.
The software used in simulations (GROMACS) contains numerous programs able to do perform different kind of analysis.
The secondary structure of peptide change during the simulation? How we can address the influence of acyl chain composition in the membrane-peptide interaction, if structural information (deuterium order parameter, bilayer thickness) from the membrane were not calculated from MD simulations?
It's possible to calculate distance and number of contacts between peptide and headgroups along the simulation and quantifies the headgroup atoms interacting with the peptide.
All this structural information can be extracted from the MD simulations and would be interesting to see in text in order to enrich the discussion and final conclusions.
Author Response
We thank the reviewer for their comments. Our point-by-point response and a marked-up version of the manuscript is attached.

Reviewer 2 Report
This study found a non-linear relationship between the interaction of the HIV-1 fusion peptide (FP) and the lipid bilayer composition when varying the acyl chain content while keeping the head group composition constant. This study was performed using circular dichroism spectroscopy, small-angle neutron scattering, neutron spin echo spectroscopy, and molecular dynamics simulations. The MD simulations showed that the area around the peptide was enhanced in the presence of POPG and depleted of DMPC, and POPC was slightly more prevalent in the vicinity of the peptide than DMPG, suggesting lipid-specific interactions in the lipid mixture. The study attributes this behavior to electrostatic interactions and reduction of curvature strain due to favorable head group interactions with POPG and POPC. However, the results of this study appear to contradict previous studies on the impact of HIV-1 FP on lipid bilayer membranes. Overall, the results showed that the peptide-bilayer interaction is not solely dependent on the unsaturated lipid acyl chain content, but rather influenced by lipid-specific interactions. The authors used a variety of experimental computational techniques to understand the interactions between HIV-1 fusion peptide with lipid bilayer vesicles. However, I have a few concerns and suggestions. Since I am not an experimentalist, my review is only focused on MD results.
1) It would be helpful if the authors perform RMSD, RMSF, lipid-protein contacts, and secondary structural analysis on the MD trajectories of the Lipid-3 system. Though it is computationally expensive, it would also be interesting to unravel the change in secondary structural conformation of gp41rk w.r.t to change in lipid bilayer composition.
2) Cholesterol has been shown to increase the fluidity of lipid bilayers. It does so by reducing the packing of the acyl chains and decreasing the rigidity of the bilayer. Cholesterol also has a tendency to accumulate at the edges of lipid domains, known as lipid rafts, which can further increase the fluidity of the bilayer. However, the effect of cholesterol on bilayer fluidity is complex and dependent on various factors, such as the concentration of cholesterol, the composition of the bilayer, and temperature. I think the authors need to focus more on this aspect and discuss their MD simulations data.
3) The choice of setting the maximum calculation distance for the gp41rk RDFs to 25 Å is not justified.
Author Response

(The authors gave the same response as above.)

Round 2
Reviewer 1 Report
Thank you for the assertive way to answer all questions. I just recommend to review the new text added to manuscript, because it's possible to see some typos.